



**Characteristics, main sources, health risks of PM$_{2.5}$-bound**
**perfluoroalkyl acids in Zhengzhou, central China: From**
**seasonal variation perspective**
**Jingshen Zhang**[1,3]**, Xibin Ma**[2]**, Minzhen Li**[2]**, Zichen Wang**[2]**, Nan Jiang**[2,1]**,**
**Fengchang Wu**[3,4]
[1]College of Chemistry, Zhengzhou University, Zhengzhou 450001, China
[2]College of Ecology and Environment, Zhengzhou University, Zhengzhou
450001, China
[3]Huang Huai Laboratory, Henan Academy of Sciences, Zhengzhou 450046,
China
[4]State Key Laboratory of Environmental Criteria and Risk Assessment, Chinese
Research Academy of Environmental Sciences, Beijing 100012, China
*Correspondence to:* Nan Jiang (jiangn@zzu.edu.cn), Xibin Ma
(maxibin163@163.com)





## Abstract

Perfluoroalkyl acids (PFAAs) have become the focus due to their physicochemical stability and potential toxicity. In this study, the investigation aimed to characterize the pollution levels, identify the primary sources, and assess the health risks associated with PFAAs in $PM_{2.5}$. The average concentration range for PFAAs were between 46.68 and 181.63 $pg \cdot m^{-3}$, with the main components being perfluorooctanoic acid (PFOA), perfluorooctane sulfonate (PFOS), and perfluorobutanoic acid. PFAA concentrations in $PM_{2.5}$ were greatly influenced by the short- and medium-range air masses, and markedly elevated by industrial activities in surrounding urban areas. The results by positive matrix factorization revealed that PFOA-based products (38.2%) and degradation byproducts of fluorotelomer alcohols (26.7%) were the predominant sources. The average daily inhalation of 17 PFAAs fluctuated greatly (median: $4.35 \times 10^{-3}$ to 8.78 $pg \cdot (kg \cdot d)^{-1}$), showing different seasonal variations with estimated daily intake of PFOA and PFOS reaching peak value in winter (5869.39 pg) and spring (4219.41 pg), respectively. The research indicated that seasonal regulation of PFOA-related manufacturing and joint pollution control with neighboring cities could reduce PFAAs levels in $PM_{2.5}$. The results provided theoretical support for government to make targeted control plans for PFAAs and basic data for relevant researchers.

**Keywords:** PFAAs, , PMF model, source apportionment, health risks.



## 1 Introduction

Perfluoroalkyl Acids (PFAAs), a subset of perfluoroalkyl substances (PFASs), can form smooth surfaces that are waterproof, oil-resistant, and stain-resistant, hence their widespread application in various industrial productions, such as paints, surfactants, coatings, emulsifiers, and fire retardants (Lindstrom et al., 2011). During the production and utilization of PFAA-containing products, PFAAs are released into a variety of environment. Consequently, PFAAs could be detected in the human body (Cardenas et al., 2017), the atmosphere, water, or snow (Dreyer et al., 2009; Hu et al., 2016; Wang et al., 2017) and wildlife (Sedlak et al., 2017). PFAAs, having environmental stability, potential for long-range transport and toxicity, cause significant risks to environment and human health (Wang et al., 2022a; Wu et al., 2022). PFAAs levels in the atmosphere have attracted adequate attention due to people breathe second by second.

The PFAAs concentration range in the atmosphere of Japan and Malaysia were 3.7–330 $pg \cdot m^{-3}$, with perfluorobutanoic acid (PFBA) exhibiting the highest concentrations (Wang et al., 2022b). The atmospheric concentration range of $\sum_{13}$PFAAs in Chinese cities was between 6.19 and 292.57 $pg \cdot m^{-3}$, with an average value of 39.84 ± 28.08 $pg \cdot m^{-3}$, exceeding the values in other countries. The predominant constituent was identified perfluorooctanoic acid (PFOA) (Han et al., 2019). PFOA and perfluorooctane sulfonate (PFOS) were the primary components of PFAAs in the atmosphere of Shenzhen, accounting for approximately 35% and 22% of PFAAs (Liu et al., 2015a). The PFAAs peak concentrations occurred during spring (97.5–709 $pg \cdot L^{-1}$), while autumn recorded the lowest levels (9.27–105 $pg \cdot L^{-1}$), exhibiting a seasonal variation in Chengdu (Fang et al., 2019). Due to their low volatility, PFAAs tend to be more prevalent in the particulate phase (Liu et al., 2018). The previous study found that most PFAAs in the atmosphere are concentrated in the particle phase rather than the gas phase, especially perfluoroalkyl carboxylic acids (PFCAs) tending to distribute in $PM_{2.5}$ (Heydebreck et al., 2016; Lin et al., 2020). $PM_{2.5}$ have the capacity to penetrate deep into the lungs, so health risks of



PM$_{2.5}$-bound PFAAs have more health risks than PFAAs alone, and the synergistic
effects of PFAAs in PM$_{2.5}$ have become a key public health priority (Qiao et al.,
2024). In a whole, there is a lack of seasonal comparative studies on PM$_{2.5}$-bound
PFAAs in densely populated inland urban areas.
PFAAs can be directly emitted into the atmosphere during production,
transportation, application, and disposal processes (Dong et al., 2021), and enter other
environment through atmospheric dry and wet deposition (Barton et al., 2006).
Studies have demonstrated that long range atmospheric transport (LRAT) is a
significant process influencing the distribution of PFAAs (Gawor et al., 2014; Jahnke
et al., 2007), serving as a key source for remote inland regions (Ellis et al., 2004;
Murr, 2020) and even polar (Wang et al., 2014). Receptor model was successfully
used in source apportionment of PFAAs. Han et al. employed positive matrix
factorization (PMF) to identify four sources of PFAAs within the atmosphere.
Meanwhile, Chen and Wang combined principal component analysis with
back-trajectory model to assess air mass influence PFAA concentrations in
precipitation from the Tibetan Plateau (Han et al., 2019) and airborne particulate
matter in Chengdu, China (Chen et al., 2021). Direct emissions associated with
fluoropolymer manufacturing and indirect contributions from incomplete degradation
of precursors are the main sources of PFAAs in the atmosphere (Barber et al., 2007).
For instance, fluorotelomer alcohols (FTOHs) are oxidized by hydroxyl radicals
leading to the formation of PFAAs (Thackray and Selin, 2017). PFAAs are known to
be carcinogenic and exposure assessments were conducted in previous studies. The
average daily inhalation (ADI) of PFOA and PFOS were quantified, ranging from
0.05–11.97 pg·(kg·d)$^{-1}$ and 0.03–8.90 pg·(kg·d)$^{-1}$, respectively (Lin et al., 2022;
Liu et al., 2015a; Liu et al., 2023; Liu et al., 2018). According to human
epidemiological studies, the European Food Safety Authority (EFSA) has
delineated a tolerable weekly intake for PFOS at 13 ng·kg$^{-1}$ and for PFOA at 6
ng·kg$^{-1}$ (Yeung et al., 2019). In brief, few studies have begun to focus on the
source and health risks of PFAAs, however no systematic studies have been
conducted of PFAAs in PM$_{2.5}$.



Given a comprehensive research of PFAAs in PM$_{2.5}$ is important for
enhancing our understanding of the environmental activity, so the pollution
characteristics, sources and health risks of PM$_{2.5}$-bound PFAAs were studied. The
PM$_{2.5}$ samples were collected in Zhengzhou, central China, characterized by dense
population (12.828 million resident population in 2022) (Statistics, 2023) and
heavy PM$_{2.5}$ pollution (47.7 μg·m$^{-3}$ in 2022, exceeding the national average by
64.5%) (Department of Ecology and Environment of Henan Province, 2022; Ministry
of Ecology and Environment of the People′s Republic of China, 2022), and 17
PFAAs were analyzed in this study. The objectives of this study were (1) to
characterize seasonal variations in PFAA pollution in PM$_{2.5}$, (2) to employ multiple
models (including back trajectory model, potential source contribution function
(PSCF) and PMF model) to identify primary sources as well as potential regional
sources contributing to PFAAs, and (3) to evaluate health risks associated with
PFAAs in PM$_{2.5}$ in four seasons. This study conducted a systemative investigation
of PM$_{2.5}$-bound PFAAs in a typical rapidly developing city with relative high
PM$_{2.5}$ pollution, providing an integrated analysis of the pollution characteristics,
source identification, and health risks of PFAAs, thereby expanding the existing
data of knowledge and providing a theoretical basis for the government to make
control plans on PFAAs in different seasons.

## 115   2 Material and methods

### 116   2.1 Sample collection

PM$_{2.5}$ samples were collected from the rooftop of the Collaborative
Innovation Building at Zhengzhou University (34°48′N, 113°31′E) on the roof (14
m height), approximately 500 meters east of the West Fourth Ring Road and 2
kilometers south of the Lianhuo Expressway. A total of 60 valid samples were
collected from Dec 2022 to Nov 2023 (details in Table S3). The diameter of the
quartz membrane was 90 mm, with sampling conducted from 10:00 to 09:00 on the



following day by using a sampler (JCH-6120-1, Ju Chuang Environmental inc.,
China) at a flow rate of 100 L/min. Before sampling, quartz filters were wrapped
in aluminum foil and baked in a muffle furnace at 450°C for 5 hours to eliminate
organic components. The quartz filters were placed in a super clean room
(temperature of 20 ± 5°C; relative humidity of 50± 5%) for 48 hours. The quartz
filters were changed daily in the ultra-clean room. Clean the instrument with
alcohol cotton before and after each sample, and record the sampler's standard
condition volume. The quartz filters were weighed twice before and after sampling,
and the error between the two times was not more than 10 mg. After weighing the
sample, the filters were wrapped in aluminum foil and stored at -18° C until the
sample was used. The samples would be deemed invalid when adverse weather
conditions (such as rain or snow) or power outages occurred during sampling
process.

## 2.2 Chemicals and reagents

The chemical reagents used in this study were 17 kinds of PFAAs mixed
standard solutions and 9 kinds of mass-labeled internal standard mixed standard
solutions. 17 PFAAs mixed standard solutions: PFBA, Perfluoropentanoic acid
(PFPeA), Perfluorohexanoic acid (PFHxA), Perfluoroheptanoic acid (PFHpA), PFOA,
Perfluorononanoic    acid    (PFNA),    Perfluorodecanoic    acid    (PFDA),
Perfluoroundecanoic acid (PFUnDA), Perfluorododecanoic acid (PFDoDA),
Perfluorotridecanoic acid (PFTrDA), Perfluorotetradecanoic acid (PFTeDA),
Perfluorohexadecanoic acid (PFHxDA), Perfluorooctadecanoic acid (PFODA),
Perfluorobutane sulfonate (PFBS), Perfluorohexane sulfonate (PFHxS), PFOS, and
Perfluorodecane sulfonate (PFDS). 9 kinds of mass-labeled internal standard mixed
solutions: $^{13}C_4PFBA$, $^{13}C_4PFHxA$, $^{13}C_4PFOA$, $^{13}C_4PFNA$, $^{13}C_4PFDA$, $^{13}C_4PFUnDA$,
$^{13}C_2PFDoDA$, $^{18}O_2PFHxS$, and $^{13}C_4PFOS$.



## 2.3 Sample preparation and instrument analysis


After the addition of methanol, the extracts were performed 3 times by
sonication. Following the centrifugation (4500 r/min, 15 min), the extracts were
diluted with ultrapure water. The extracts were purified using weak anion exchange
cartridges and then concentrated to 200 μL with nitrogen. Prior to instrumental
analysis, the sample was filtered through a 0.22 μm nylon membrane and
transferred into a 2 mL brown injection vial. Detailed steps for sample pretreatment
are documented in supplementary 1.2.
The analysis of PFAAs was performed using Ultra High Performance Liquid
Chromatography-Tandem Mass Spectrometry (Ekspcrt nano Lc425, Singapore)
UPLC-MS/MS. The analytical instrument employed consisted of a triple
quadrupole liquid chromatography-mass spectrometer. For chromatographic
separation, a $C_{18}$ reverse-phase column (150 mm × 2.1 mm, 1.8 μm) was selected.
Comprehensive details regarding the instrumental analysis can be found in
supplementary 1.2.

## 2.4 Quality assurance and quality control


During the sample collection, processing, and analysis phases, fluorinated
plastic materials were avoided, such as polytetrafluoroethylene (PTFE). Use
ceramic scissors to cut quartz filters and wipe the scissor with methanol before
cutting another sample to avoid excess particles affecting the next sample. The
polypropylene tubes were used. All samplers and containers were precleaned with
methanol. The concentrations of the prepared 7-point calibration solution were as
follows 0.1, 1, 5, 10, 50, 100, and 200 μg/L. The concentration of internal standard
solution was 10 ng/mL. The procedure blanks were prepared using the same
methods as the samples. Two field blank membranes were collected during each
seasonal sampling period. The final concentrations of PFAAs were determined by
subtracting the concentrations of the procedure blanks from those of the samples.



Reagent blanks were employed to monitor instrumentation performance. No PFAAs were detected in field blanks and program blanks. The method detection limit (MDL) was calculated based on three times the standard deviation of the blanks. If PFAAs were not detected in the blanks, MDL refers to a concentration corresponding to peak intensity with a signal-to-noise ratio (S/N) of 3. Values below MDL were replaced with half of MDL (Han et al., 2019; Li et al., 2024). The MDL value and Mark recovery ranged from 0.2–0.3 (ng/L) and 71.27%–118.08% respectively. Detailed information on the individual compounds of PFAAs is documented in Table S1.

The PMF model was used to cluster PFAAs with similar sources to identify potential sources. The ADI model was employed to quantitatively evaluate the health risks posed by PFAAs to human populations. The detailed information of PMF and ADI models could be found in supplementary 1.2 and 1.3, which provides an in-depth explanation of these analytical frameworks.

## 3 Results and discussion

### 3.1 Characteristics of PFAAs in PM$_{2.5}$

The seasonal average concentrations ranged from 46.68 to 181.63 pg·m$^{-3}$ in Fig. 1, which was comparable to levels observed in Chengdu (150 pg·m$^{-3}$) (Fang et al., 2019), but significantly higher than those recorded in Shenzhen (8.80 pg·m$^{-3}$) (Liu et al., 2015a) and the average concentration in China (39.84 pg·m$^{-3}$) (Han et al., 2019). These factors, which characterized this region as having a dense population, concentrated industrial activities, and serious PM$_{2.5}$ pollution, may contribute to higher PFAA levels than other cities. As shown in Fig. 2, the PFAA concentrations in PM$_{2.5}$ peaked during winter and were 1.7 times higher than autumn level and 3.9 times higher than summer level. The result indicated that PFAAs had obvious seasonal variation. The long-chain PFAA concentrations (1169.60 pg·m$^{-3}$) significantly exceeded that of short-chain PFAAs (915.24 pg·m$^{-}$





$^3$), consistent with the findings in researches (Han et al., 2019; Tian et al., 2018).
Detection rates for PFOA, PFPeA, and PFBA in four seasons reached 100%, while
detection rates for PFHxA, PFHpA, PFBS, and PFOS exceeded 80%. During the
study period, PFOA and PFOS along with its primary substitutes accounted for
23%–34% and 18.1%–29.9% of total PFAAs, consistent with the research (Liu et al.,

208   2017).

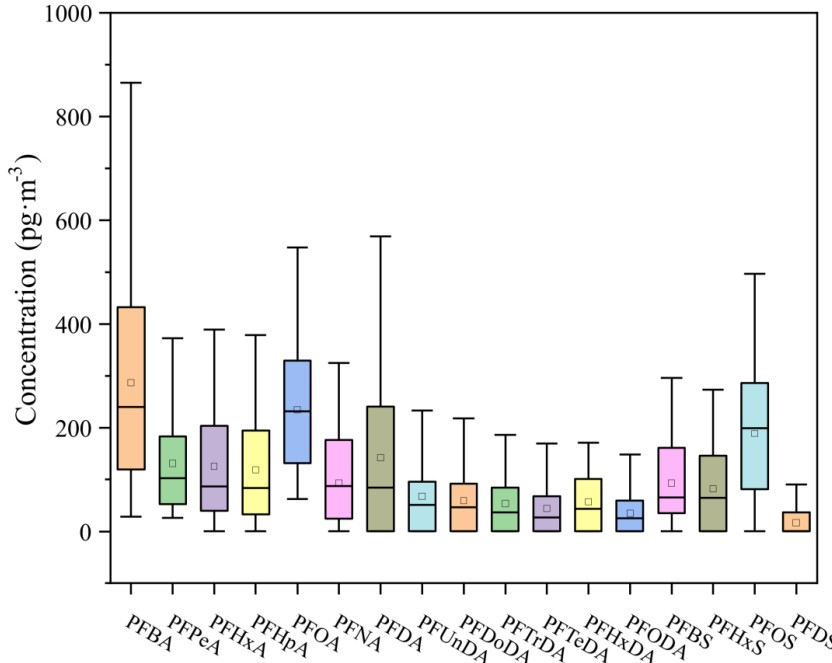

209                         Fig. 1. Box diagram of PFAAs concentrations in PM$_{2.5}$

210        The content of PFOA and its substitutes reached 23% of total PFAAs in

autumn, 34% in winter and, and 31% in spring and summer. Their applications
span across the chemical industry and domestic activities, particularly in the
manufacture of plastic and rubber commodities (Liu et al., 2015a; Prevedouros et
al., 2006). The rising domestic demand and industrial output of PFOA products
were outstanding trends within China (Du et al., 2023). The mean concentration of
PFOA (294.52 ± 215.40 pg·m$^{-3}$) in Zhengzhou markedly surpassed those recorded
in Chengdu (42.3±54.4 pg·m$^{-3}$), Ireland (8.9 pg·m$^{-3}$), and Japan (Tsukuba, 2.6





pg·m$^{-3}$; Morioka, 2.0 pg·m$^{-3}$), but it fell below the levels detected in Changshu,
China (556.0 pg·m$^{-3}$), a local area of fluorochemical industrial park (Barber et al.,
2007; Fang et al., 2019; Harada et al., 2005; Yu et al., 2018). The content of PFOS
and its substitutes were more than 25% in winter and summer, more than 20% in
autumn, and more than 10% in spring. PFOS is extensively utilized in metal
electroplating, firefighting foams, the semiconductor industry, paper treatment,
textiles, and leather processing (Liu et al., 2017). PFPeA and PFBS are the
principal substitutes to long-chain PFAAs in China, being emitted during the
production of PFOS products (Liu et al., 2017). The researches have identified
PFHxDA as a degradation byproduct of substances based on FTOHs (Ellis et al.,
2004; Loewen et al., 2005). The PFHxDA concentration escalated from 2.2% in
winter to 10.4% in spring, potentially attributable to enhanced atmospheric
oxidation.

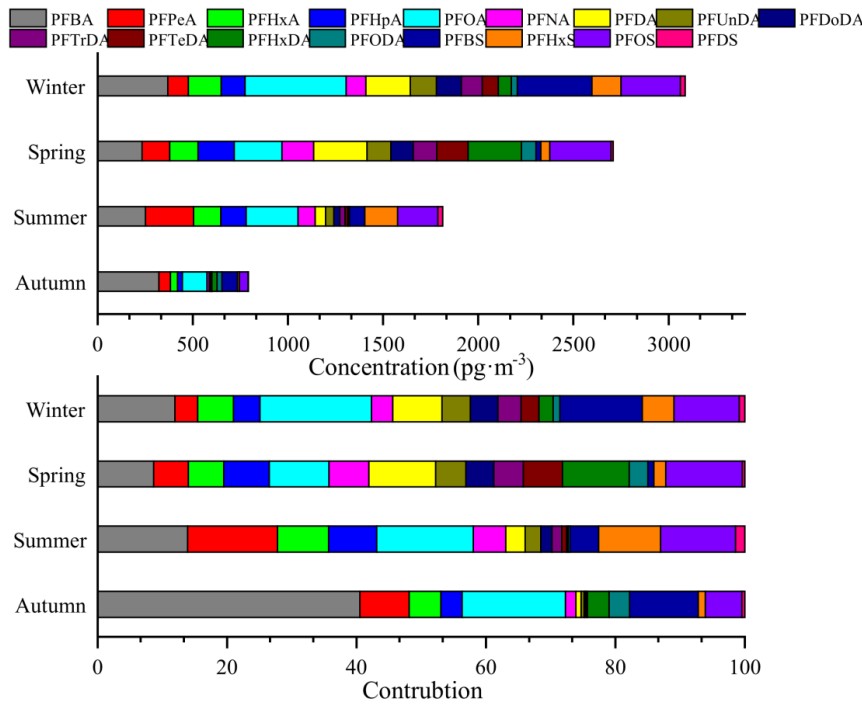

231                Fig. 2. PFAAs concentrations characteristics across four seasons

232        The seasonal distribution of PFAAs in the study region exhibited a pattern



where winter concentrations surpassed those of other seasons, with the lowest in
autumn. This seasonal variation correlated with the heightened $PM_{2.5}$ pollution
during the winter in this region. It was noteworthy that PFAA concentrations
during autumn (46.68 pg·m$^{-3}$), when were at the minimum, still exceeded the
national average concentration of 39.84 pg·m$^{-3}$ (Han et al., 2019). The chemical
industry and domestic activities were the primary contributors to PFAAs pollution
in this region. A comprehensive analysis of the pollution characteristics and
sources of PFAAs in $PM_{2.5}$ was important for generating strategies aiming at
release PFAAs pollution.

## 242    3.2 Analysis of potential regional sources of PFAAs in $PM_{2.5}$

The content of PFAAs in the atmosphere is easily influenced by the transport of
atmospheric air masses (Liu et al., 2015a). As shown in Fig. 3, this study conducted a
meteorological trajectory cluster analysis during the sampling period. The spring
season was most influenced by short-range atmospheric air masses (accounting for
40.4%) in this study region. The air mass originated from Hubei Province, passed
through Middle-Lower Yangtze River plains (0.26–1.90 pg·m$^{-3}$) and then entered the
study region (Faust, 2023). This air mass would reduce the content of PFAA
concentrations in the study area because of the slow diffusion of pollutants caused by
relative stability of this air mass and the lower PFAA concentrations than this region.
The study region was also affected by the transport of long-range air masses from the
northwest direction (accounting for 38.5%), which passed through the Inner Mongolia
and Loess Plateau and the Taihang Mountains. In the autumn, the study region was
more influenced by long-range air masses from the northwest (accounting for 57.7%),
which passed through Inner Mongolia and the Loess Plateau to reach the study area.
In winter, all trajectory clusters, accounting for 10.0%, 23.3%, and 66.7% respectively,
originated from the northwest, indicating a pronounced influence of the cold air from
that direction. The increased use of urban coal combustion in winter along this
direction tended to create polluted air masses, which were then transported and



increased the pollution levels in the study region by northwesterly winds. The
long-range air masses, passing through the Inner Mongolia Plateau and the Loess
Plateau of northwest, generated the most important influence on the seasonal transport
patterns during summer, autumn, and winter in the study region. Northwest China is
situated in a plateau region. The high-altitude region has a cold-trapping effect on
PFAAs in the atmosphere (Gouin et al., 2004), which can effectively reduce the
content of PFAAs in atmospheric air masses. The Loess Plateau could weaken the
influence of air masses from the northwest on PM$_{2.5}$-bound PFAAs levels in the study
region. This result was consistent with the analysis of potential sources of PFAAs
using the PSCF below.

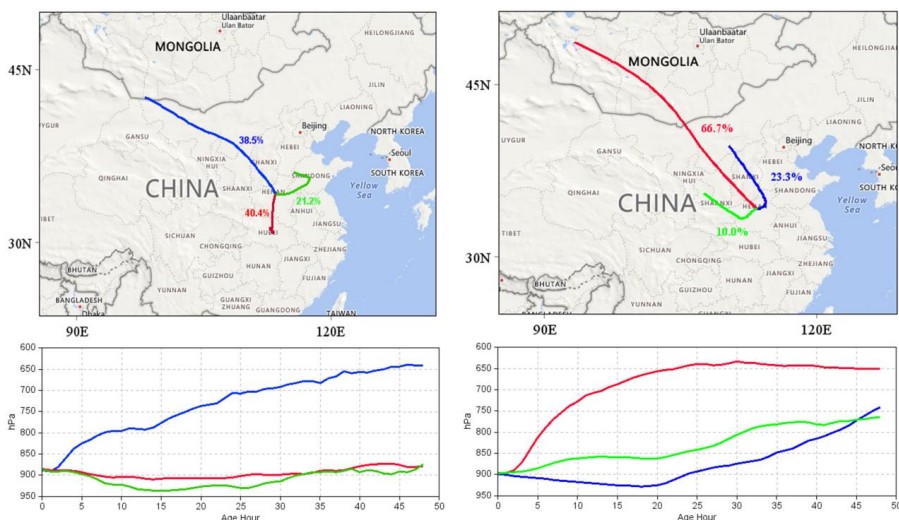

Fig. 3. Cluster analysis map of backward trajectories in Zhengzhou City (left and right are spring
and winter respectively, created by MeteoInfoMap 3.5.11 (Wang, 2014; Wang, 2019)). ©
Microsoft. The software is open.

274        As shown in Fig. 4, the strong potential source regions (WPSCF ≥ 0.5) during

spring were primarily concentrated in the southwest region of the province,
significantly influenced by the southwest air masses. The strong potential source
regions were mainly found in the surrounding regions of the province during summer,
transferred to the northwest and southwest outside the province during autumn, and
were principally distributed in the northwest and northeast within the province during



winter. The study results indicated that the influence of the northeast air masses on the
distribution of potential sources was more obvious in the study region. This result
could be attributed to the fact that the study region was located in the typical
industrial province characterized by industries such as textile treatment, metal
electroplating, and fire-fighting foam manufacturing. The wide use of PFAAs in
industrial production, such as emulsifiers and fluoropolymers, had led to increased
emissions of these substances into the atmosphere. Additionally, human activities,
such as the use of non-stick coatings on cookware and waterproof and stain-resistant
materials, particularly in densely populated areas near study region, heightened
PFAAs pollution levels (Dewapriya et al., 2023; Dhore and Murthy, 2021; Grunfeld et
al., 2024; Li et al., 2024; Wang et al., 2024). This result was consistent with
conclusions drawn by Chen et al. (Han et al., 2019). Seasonal variation could cause
the distribution of strong potential source regions to change. In contrast to spring and
summer, the distribution of strong potential source regions were more influenced by
the northwest air masses in autumn and winter. In addition to autumn, strong potential
source regions mainly distributed in the surrounding regions of the province in spring,
summer and winter.
Research indicated that the PFAAs levels in $PM_{2.5}$ were more influence by
medium- and short-range air masses and terrain. To control PFAAs levels in $PM_{2.5}$, it
is necessary to not only manage local emissions but also identify the pollution
transport pathways and sources across different seasons. Strengthen the joint
prevention and control of neighboring cities on a seasonal basis. The results of this
research provided a theoretical basis for the formulation of policies related to the
control of PFAAs levels in $PM_{2.5}$.



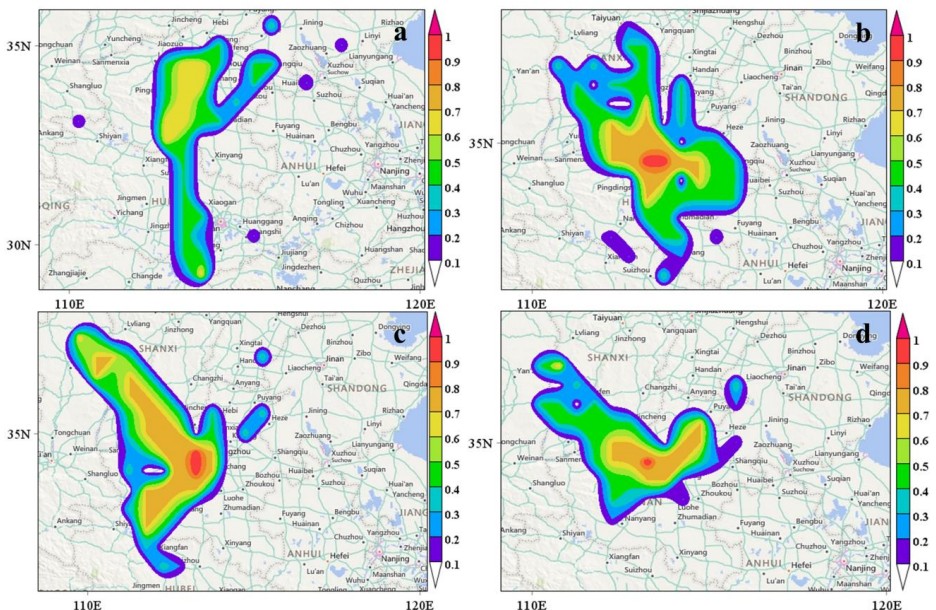

Fig. 4. Map of potential source analysis of PM$_{2.5}$-bound PFAAs in Zhengzhou City in four seasons (a, b, c and d are spring, summer, fall and winter respectively, created by MeteoInfoMap 3.5.11 (Wang, 2014; Wang, 2019)). © Microsoft. The software is open.

## 3.3 PMF receptor analysis

To further investigate the potential PFAAs sources in PM$_{2.5}$, this study employed PMF for source apportionment of PFAAs. As illustrated in Fig. 5(b), Factor 1 was predominantly characterized by high loadings of PFUnDA (72.5%), PFDoDA (71.4%), PFTrDA (80.4%), and PFTeDA (96.0%). Long-chain PFAAs (C11–C14) were known degradation products of FTOHs (Liu et al., 2017; Thackray and Selin, 2017; Wang et al., 2014). The global accumulated estimates for PFUdA, PFDoDA, PFTrDA, and PFTeDA ranged from 9 to 230 tons from 2003 to 2015, and the research shown an expected release of between 0 to 84 tons from 2016 to 2030 based on the lifecycle use and emission patterns associated with fluorocomplexes and other fluorine-containing products (Wang et al., 2014). Therefore, this factor, contributing 26.7% to total PFAAs, was thought to be the degradation products of FTOHs.

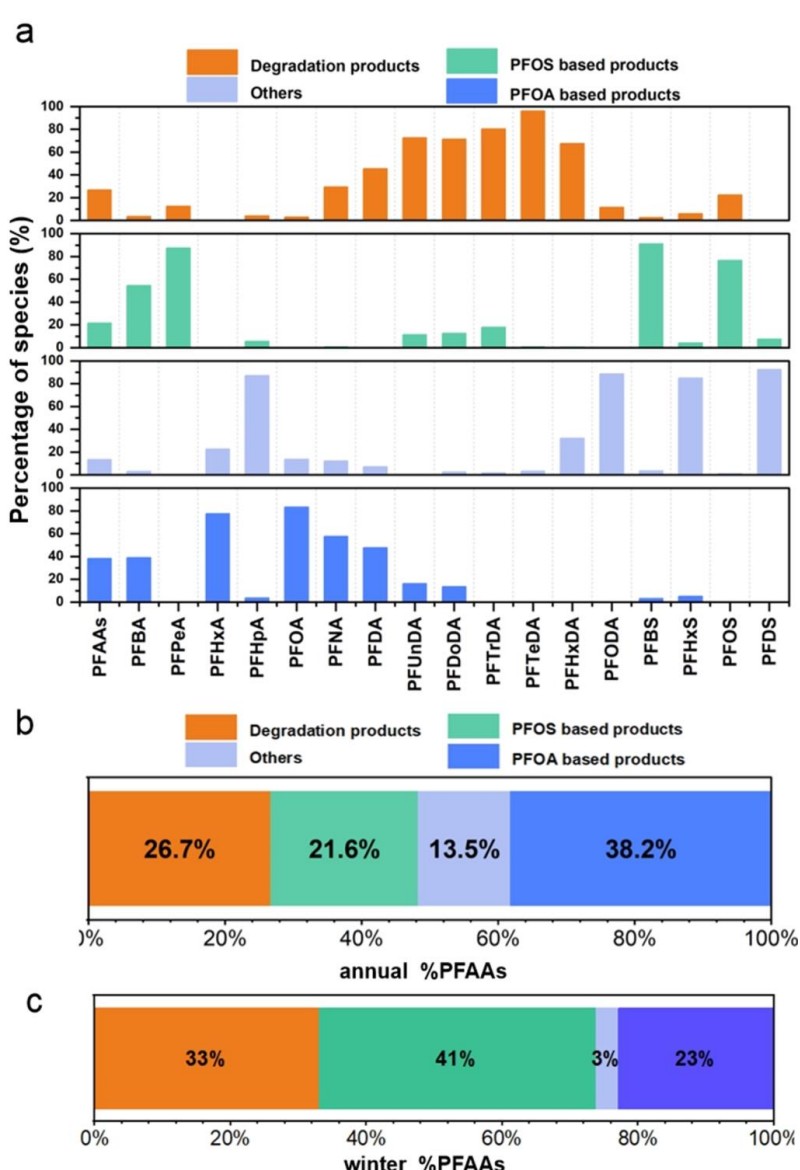

Fig. 5. The source distribution spectrum of PFAAs in PMF (a) and the source proportion diagram (b)

According to the PMF analysis results, it indicated that PFPeA, PFBS, and PFOS may originate from a common source with contribution rates of 87.4%, 91.0%, and 76.6% in Factor 2 respectively. The research indicated hat three primary kinds of chemicals related to PFOS-namely perfluorooctane sulfonates, substances containing





these compounds and polymers were widely useded in industrial production (Xie et
al., 2013). The use of PFOS-related products have resulted in the emission of PFOS
into the atmosphere during both industrial processes and human activities. PFPeA and
PFBS, because of being the significant substitutes of long-chain PFAAs, may be
released as impurities or by-products during the manufacturing of PFOS-based
products (Liu et al., 2017). Therefore, this factor, contributing 21.6% to total PFAAs,
was regarded as a direct source of perfluorooctane sulfonic acid products.
Factor 3 was characterized by high loadings of PFHpA (loading value: 87.1%)
and PFHxS (loading value: 85.0%). The formation and transformation for PFHpA and
its derivatives remained unclear. The factor containing only PFHxS did not point to a
specific source. Therefore, it was thought to be other sources influenced potentially by
atmospheric air masses or alternative origins. Thus, the contribution from the source
was thought to be 13.5% for PFAAs.
Factor 4 was identified as the primary source of PFOA products manufacturing,
characterized by significant loadings of PFHxA (77.5%), PFOA (83.4%), PFNA
(77.5%), and PFDA (47.6%). PFOA had been widely used as an emulsifying agent in
the production of plastics, rubber products, textile flame retardants, paper surface
treatments, fire-fighting foams, and PTFE emulsifiers (Liu et al., 2015b). The research
indicated that due to a rapid increase in domestic demand for PFOA products in China,
the emissions of PFCAs from factories producing these substances have increased
(Wang et al., 2014). PFOA, PFNA and their substitutes could be released through
waste gases. The contribution of this source to PFAAs accounted for 38.2%.
The sources of PFAAs are multifaceted and seasonal. Source apportionment was
conducted in winter when PFAAs pollution was most severe. As shown in Fig. 5(c),
PFOS products contributed the most to PFAAs sources in winter $PM_{2.5}$ (41%),
followed by FTOHs degradation products (33%). Factor analysis indicated the
contributions of PFAAs in $PM_{2.5}$ came from the degradation of specific fluorinated
products and direct emissions from industrial productions. The analysis of long-chain
PFAAs emphasized the potential environmental impact associated with the production
and use of FTOHs with degradation products contributing 26.7% to PFAAs in $PM_{2.5}$.



Furthermore, contributions from PFOS- and PFOA-related compounds to PFAAs in
$PM_{2.5}$ were found to be 21.6% and 38.2%. Additionally, it was thought that 13.5% of
PFAAs originated from unknown sources, and indicated a significant gap in our
understanding regarding their environmental behavior. This finding emphasizes the
urgent need for further research aiming at enhancing our comprehension of PFAAs in
$PM_{2.5}$.

## 361 3.4 Environmental indication of health impact risk

Fig. 6 illustrated the ADI of PFAAs in $PM_{2.5}$. The median ADI ranged from
$4.35 \times 10^{-3}$ to 8.78 pg·$(kg·d)^{-1}$, with relative high values for PFBA, PFOA, and
PFOS in four seasons. Notably, PFOA exhibited a median ADI as high as 8.78
pg·$(kg·d)^{-1}$, with potential carcinogenicity risk on human immune and reproductive
systems (Lin et al., 2022). The high ADI values of these compounds raise concerns
regarding their potential health impacts, especially given that $PM_{2.5}$ can be inhaled
into human lungs, thereby complicating the health implications of exposure to
$PM_{2.5}$ containing PFAAs. Although the ADI levels of these compounds remained
below the tolerable intake limits set by the EFSA (Yeung et al., 2019), it is
important to consider that PFAAs are resistant to degradation within the human
body. For example PFOS has a half-life of approximately 5.4 years (Wei et al.,
2023). Therefore, long-term exposure to lower concentrations of PFAAs than limit
values still may accumulate over time and potentially lead to adverse health
outcomes. This study discovered pronounced seasonal variation in the estimated
daily intake (EDI) (Fig. 7). The PFOA and PFOS EDI exhibited the remarkable
peak during winter (the median values: 5869.39 pg) and spring (the median values:
4219.41 pg) respectively, and recorded the lowest average daily exposure dose
during autumn (the median values 1787.21 and 3285.28 pg). A comparative
analysis of the seasonal EDI patterns indicated that the winter season was
characterized by a relatively elevated daily exposure dose, particularly for PFOA.
The observed seasonal fluctuations in EDI were due to changes in concentration



383 due to a combination of influence factors such as ambient temperature, relative

384 humidity, human activities, and atmospheric air mass transport. For example, these

385 factors comprehensively influenced the atmospheric partitioning and deposition of

386 PFOA and PFOS, thereby impacting the population's exposure to these PFAAs.

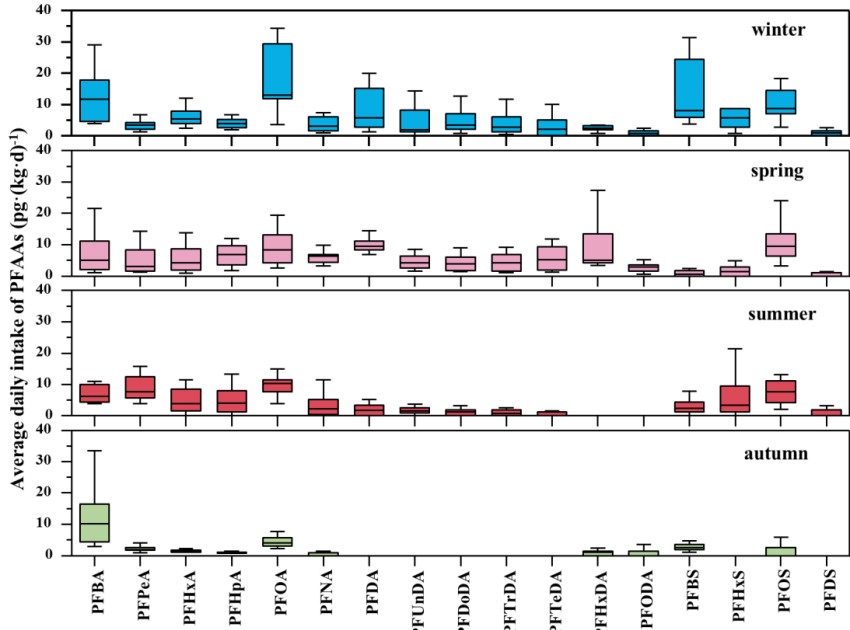

387     Fig. 6. Median Average Daily Intake (ADI) of PFAAs in PM$_{2.5}$ in four seasons


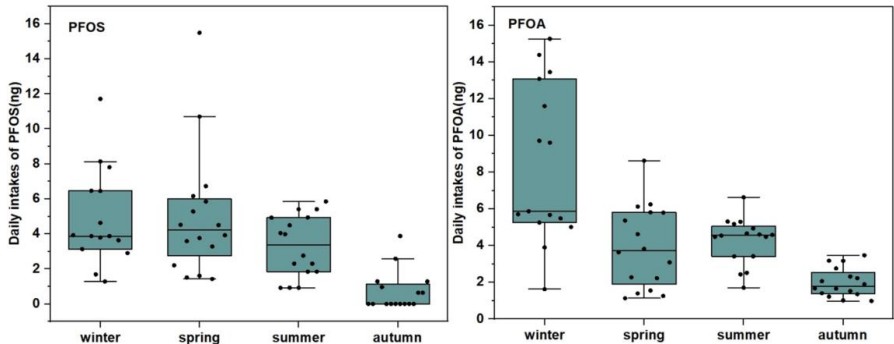

Fig. 7. The median daily estimated intake (EDI) of PFOA and PFOS varies by season in $PM_{2.5}$
To mitigate PFAAs contamination and protect environmental health, it is
recommended to strengthen regulatory controls on industrial emissions, upgrade
wastewater treatment technologies, and enhance public awareness of PFAAs risks.
Regular monitoring of PFAAs in environmental media is crucial, and industries
should be incentivized to adopt safer alternatives. Additionally, further research is
needed to better understand the long-term environmental and health impacts of
PFAAs exposure.

## 4 Conclusion

This study conducted a one-year sampling of $PM_{2.5}$ and utilized UPLC-MS/MS
to detect PFAAs in the samples. A comprehensive analysis of the pollution
characteristics, source apportionment, and health risk assessment of PFAAs in $PM_{2.5}$
was conducted. The results indicated that the detection rates of PFOA, PFPeA and
PFBA were 100%, PFHxA, PFHpA, PFBS and PFOS were more than 80%. PFAA
concentrations were highest in winter (mean value: 181.63 $pg \cdot m^{-3}$) and lowest in
autumn (mean value: 46.68 $pg \cdot m^{-3}$), however the lowest values still significantly
higher than the national average from previous study. PFOA and PFOS along with its
substitutes were primary PFAAs in $PM_{2.5}$. Backward trajectory analysis of the study
region revealed that the PFAA concentrations were susceptible to medium and



short-range atmospheric air mass transport. Controlling the concentration of PFAAs in
$PM_{2.5}$ requires primarily reducing local emissions and strengthening joint prevention
in different seasons. PMF analysis indicated that the main PFAAs sources were
products of PFOA and its substitutes (38.2%), degradation products of
fluorotelomer-based products (26.7%) and PFOS and its substitutes (21.6%). There
was also an unknown source accounting for 13.6%, indicating that there are still
significant limitations in our understanding of the PFAAs environmental behavior,
and further research is necessary. The PFAAs ADI was below the tolerable intake
limit set by the EFSA. The high EDI PFAAs values, which could be inhaled into
human lungs through $PM_{2.5}$, should be a concern due to their potential to complicate
health effects, making PFAAs research particularly important in regions with heavy
$PM_{2.5}$ pollution. Monitoring the impact of atmospheric air mass transport in the study
region by season, strengthening targeted joint prevention and control with
neighboring cities are crucial steps in reducing the concentration of PFAAs in $PM_{2.5}$.
The study results of concentration characteristics, origin and health effects of PFAAs
could provide theoretical support and basic data for government and follow-up
researchers to reduce PFAAs levels.



## Data availability

All raw data can be provided by the corresponding authors upon request.

## Author contributions

JZ: Writing-Review and Editing; XM: Writing-Original draft preparation, Writing-Review and Editing; ML: Writing-Review and Editing; ZW: Writing-Review and Editing; NJ: Writing-Review and Editing, Supervision, Project administration; FW: Resources.

## Competing interests

The authors declare that they have no conflict of interest.

## Acknowledgments

Funding: This research has been supported by the National Natural Science Foundation of China [Grant No. 52170117].



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
