# Peer review of "Characteristics, main sources, health risks of PM2.5-bound"

_EGUsphere, 2024_

## Author Comment (AC1)

**Itemized Response to Reviewer's Comments**

**Ms. Ref. No.**: egusphere-2024-4147

**Title:** Characteristics, main sources, health risks of PM$_{2.5}$-bound perfluoroalkyl acids in Zhengzhou, central China: From seasonal variation perspective

**RESPONSE TO REVIEWERS**

**Reviewer Comments:**

RESPONSE: We sincerely thank the valuable and constructive inputs of the reviewer on our manuscript. We believe that we have adequately addressed all comments and thus the current version has been greatly improved with those valuable comments. In the revised manuscript, all the modifications were highlighted in red.

**General Comments:**

While the data presented represent a tremendous undertaking in terms of samples collected, quality assurance, and depth of analysis, the manuscript readability could greatly benefit from an in-depth editorial review. There are numerous instances where it is either unclear or ambiguous what the authors are trying to convey (ex. lines 48 – 49 "PFAAs levels in the atmosphere have attracted adequate attention due to people breathe second by second." the phrase "systemative investigation" in line 109, and "The polypropylene tubes were used." in lines 168 – 169.) Analytically speaking, it is evident that the authors have taken careful consideration in their sampling and data analysis, with numerous blanks, quality assurance checks, and data validation methods. It is also appreciated that methods for calculating MDLs were provided, and gives evidence that the researchers were thorough in their investigation.

Response: Thank you for your valuable and constructive comments. Detailed revisions have been listed below.

**Specific Comments:**

**Comment 1:** In Figure 1, it is unclear whether the reported data is averaged across all seasons or is for a single season. Line 192 suggests the highest average concentration was 181.63 pg m$^{-3}$, but PFBA, PFOA, and PFOS all appear to have concentrations above 200 pg m$^{-3}$ in the figure. Please clarify.

Response: The data in the manuscript (46.68 – 181.63 pg·m$^{-3}$) refers to the average PFAA concentrations for each season and the data in Figure 1 refers to the PFAA concentrations for the four seasons. These sentences have been rephrased in this manuscript.

Lines 193 – 201 (New Version): The PFAA average concentrations ranged from 46.68 to 181.63 pg·m$^{-3}$ in Fig. 1 across four seasons. However, the increased airflow during pump operation enhanced the adsorption of gaseous PFAA on quartz filters (Turpin et al., 1994; McMurdo et al., 2008; Ahrens et al., 2012; Chang et al., 2024), which may lead to a slight overestimated of PFAA values in this study. The PFAA average concentrations were comparable to levels observed in Chengdu (150 pg·m$^{-3}$) (Fang et al., 2019), but significantly higher than those

recorded in Shenzhen (8.80 pg·m⁻³) (Liu et al., 2015a) and the average concentration in China (39.84 pg·m⁻³) (Han et al., 2019).

Line 215 (New Version): Fig. 1. Box diagram of 17 PFAA concentrations in PM$_{2.5}$ across four seasons.

Reference:

Fang, S., Li, C., Zhu, L., Yin, H., Yang, Y., Ye, Z., et al., 2019. Spatiotemporal distribution and isomer profiles of perfluoroalkyl acids in airborne particulate matter in Chengdu City, China. Sci. Total. Environ. 689, 1235-1243. http://dx.doi.org/10.1016/j.scitotenv.2019.06.498

Han, D., Ma, Y., Huang, C., Zhang, X., Xu, H., Zhou, Y., et al., 2019. Occurrence and source apportionment of perfluoroalkyl acids (PFAAs) in the atmosphere in China. Atmos. Chem. Phys. 19, 14107-14117. http://dx.doi.org/10.5194/acp-19-14107-2019

Liu, B., Zhang, H., Yao, D., Li, J., Xie, L., Wang, X., et al., 2015a. Perfluorinated compounds (PFCs) in the atmosphere of Shenzhen, China: Spatial distribution, sources and health risk assessment. Chemos. 138, 511-518. http://dx.doi.org/10.1016/j.chemosphere.2015.07.012

**Comment 2:** In Figure 2, it may be appropriate to increase the spacing between the text and the markers in the legend, if possible, to increase the readability of the figure.

Response: The spacing between the text and the markers in the legend has been increased to improve readability.

[Figure]

Fig. 2. PFAA concentrations characteristics across four seasons

**Comment 3:** While there are several mentions that this information can be used

to better shape policy to control PFAA levels in PM$_{2.5}$, it is unclear what such policies could entail. Lines 298 – 303 suggest that management of local emissions is one way to control PFAAs, but it would be helpful to provide potential methods for emission reductions in industrial applications.

Response: In this manuscript suggestions for reducing PFAA pollution levels have been added, with the perspectives of PFAA proportion and air mass transport pathways.

Lines 242 – 245 (New Version): Long-chain PFAAs (e.g., PFOA and PFOS) were major pollutants and require replacement with short-chain alternatives (e.g., PFBS and PFPeA) or non-fluorinated substitutes such as silicon-based emulsifiers.

Lines 309 – 314 (New Version): For example, regulate PFAA emissions from textile and electroplating industries along southern urban in spring, collaborate with northwestern provinces to curb coal combustion in key transport cities in winter, establish pollution-blocking monitoring networks at northwestern entry points (e.g., Jiaozuo city and Jiyuan city) and leveraging the Taihang Mountains and Loess Plateau to intercept pollutants, in summer and autumn.

**Comment 4:** In line 159 of the supplement, "ED" is referred to as the burst time. It may be more appropriate to refer to this as the "exposure duration". I am assuming the authors use 72 as a general value for life expectancy. Also in line 159, the EF, exposure frequency is 350 days/year. Is this to suggest a two-week annual vacation? If so, perhaps it would be beneficial to more explicitly list the assumptions made, and why.

Response: The "ED" has been amended and more detailed information of EF has been added in this supplement.

Lines 158 – 160 (New Version): EF is the annual exposure frequency (350 days·year$^{-1}$, without the time of two-week annual vacation), ED is exposure duration (72 a).

**Comment 5:** In line 226, "the researches have identified PFHxDA as…" is unclear. Perhaps "Previous studies have identified PFHxDA as…" would be better-suited here.

Response: This sentence has been rephrased.

Lines 232 – 234 (New Version): Previous studies have identified PFHxDA as a degradation byproduct of substances based on FTOHs (Ellis et al., 2004; Loewen et al., 2005).

Reference:

Ellis, D.A., Martin, J.W., De Silva, A.O., Mabury, S.A., Hurley, M.D., Andersen, M.P.S., et al., 2004. Degradation of fluorotelomer alcohols: A likely atmospheric source of perfluorinated carboxylic acids. Envion. Sci. Technol. 38, 3316-3321. http://dx.doi.org/10.1021/es049860w

Loewen, M., Halldorson, T., Wang, F.Y., Tomy, G., 2005. Fluorotelomer carboxylic acids and PFOS in rainwater from an urban center in Canada. Envion. Sci. Technol. 39, 2944-2951. http://dx.doi.org/10.1021/es048635b

**Comment 6:** In line 323, "The research indicated hat" should be "the research indicated that"

Response: This sentence has been revised.

Lines 336 – 339 (New Version): The research indicated that three primary kinds of chemicals related to PFOS-namely perfluorooctane sulfonates, substances containing these compounds and polymers were widely useded in industrial production (Xie et al., 2013).

Reference:

Xie, S., Wang, T., Liu, S., Jones, K.C., Sweetman, A.J., Lu, Y., 2013. Industrial source identification and emission estimation of perfluorooctane sulfonate n hina. Environ. Int. 52, 1-8. http://dx.doi.org/10.1016/j.envint.2012.11.004

---

## Author Comment (AC2)

1 **Itemized Response to Reviewer's Comments**

2 **Ms. Ref. No.**: egusphere-2024-4147

3 **Title:** Characteristics, main sources, health risks of $PM_{2.5}$-bound perfluoroalkyl acids

4 in Zhengzhou, central China: From seasonal variation perspective

5 RESPONSE TO REVIEWERS

6 **Reviewer Comments:**

7 RESPONSE: We sincerely thank the valuable and constructive inputs of the reviewer

8 on our manuscript. We believe that we have adequately addressed all comments and

9 thus the current version has been greatly improved with those valuable comments. In

10 the revised manuscript, all the modifications were highlighted in red.

**General comments:**

The manuscript presents a study on PM$_{2.5}$-bound perfluoroalkyl acids (PFAAs) in Zhengzhou, China, focusing on seasonal variations, source apportionment using positive matrix factorization (PMF), and health risk assessment. The study is relevant and contributes to understanding PFAS contamination in urban air. The manuscript is well-structured and logically organised. However, it contains several typographical and grammatical errors. While some of these have been highlighted in the 'technical corrections' section, the list is not exhaustive. A thorough proofreading is recommended to enhance clarity and readability.

Response: Thank you for your valuable and constructive comments. Detailed revisions have been listed below.

**Specific comments:**

**Comment 1:** L.38 – Please change the expansion of PFAS to per- and polyfluoroalkyl substances from perfluoroalkyl substances.

Response: This sentence has been rephrased.

Lines 37 – 41 (New Version): Perfluoroalkyl Acids (PFAAs), a subset of per- and polyfluoroalkyl substances (PFASs), can form smooth surfaces that are waterproof, oil-resistant, and stain-resistant, hence their widespread application in various industrial productions, such as paints, surfactants, coatings, emulsifiers, and fire retardants (Lindstrom et al., 2011).

Reference:

Lindstrom, A.B., Strynar, M.J., Libelo, E.L., 2011. Polyfluorinated Compounds: Past, Present, and Future. Envion. Sci. Technol. 45, 7954-7961. http://dx.doi.org/10.1021/es2011622

**Comment 2:** Section 2.1 – The use of quartz fiber filters during PM sampling is known to produce positive sampling artefacts such as the adsorption of gas phase compounds onto the filter. For further details on such artefacts, refer to Turpin et al. (1994) (https://doi.org/10.1016/1352-2310(94)00133-6) and Chang et al. (2024) (https://doi.org/10.1039/D4EM00359D). PFAS such as PFOA are known to partition out into gas phase from the aerosols (please see studies by Ahrens et al. 2012 (https://doi.org/10.1021/es300898s) and McMurdo et al. 2008

42  (https://doi.org/10.1021/es7032026). Additionally, short-chain PFAS, including PFBA

43  and PFBS, are semi-volatile and may exist in both the gaseous and particulate phases.

44  As a result, the PFAA concentrations measured in this study may be slightly

45  overestimated due to the potential inclusion of gaseous PFAA. Consider addressing

46  these sampling artifacts in the methodology section.

47      Response: We sincerely appreciate the reviewer for highlighting this comment. It

48  should be noted that the positive sampling artefacts raised by reviewer is existent.

49  However, there is still a lack of effective methods for PFAA sampling in particulate

50  matter and the quartz fiber filters were also widely to sample particulate matter in

51  previous researches (Fang et al., 2019; Wu et al., 2019 and Li et al., 2024). Before

52  sampling, quartz filters could be baked to remove disturb from organic matter. In

53  process blanks of this study, PFAA levels were below the method detection limits

54  (MDLs). Considering that the increased airflow during pump operation enhances the

55  adsorption of gaseous PFAA on quartz filters, this study illustrated in the Results and

56  Discussion section that the adsorption effect of quartz filters may lead to a slight

57  overestimated of PFAA concentrations.

58  Lines 194 – 197 (New Version): However, the increased airflow during pump

59  operation enhanced the adsorption of gaseous PFAA on quartz filters (Turpin et al.,

60  1994; McMurdo et al., 2008; Ahrens et al., 2012; Chang et al., 2024), which may

61  lead to a slight overestimated of PFAA values in this study.

62      Reference:

63  Ahrens, L., Harner, T., Shoeib, M., Lane, D.A., Murphy, J.G., 2012. Improved Characterization of
64      Gas-Particle Partitioning for Per- and Polyfluoroalkyl Substances in the Atmosphere Using
65      Annular Diffusion Denuder Samplers. Envion. Sci. Technol. 46, 7199-7206.
66      http://dx.doi.org/10.1021/es300898s
67  Chang, N.Y., Eichler, C.M.A., Amparo, D.E., Zhou, J., Baumann, K., Hubal, E.A.C., et al., 2024.
68      Indoor air concentrations of $PM_{2.5}$ quartz fiber filter-collected ionic PFAS and emissions to
69      outdoor air: findings from the IPA campaign. Environ. Sci.-Process Impacts.
70      http://dx.doi.org/10.1039/d4em00359d
71  Fang, S., Li, C., Zhu, L., Yin, H., Yang, Y., Ye, Z., et al., 2019. Spatiotemporal distribution and
72      isomer profiles of perfluoroalkyl acids in airborne particulate matter in Chengdu City, China.
73      Sci. Total. Environ. 689, 1235-1243. http://dx.doi.org/10.1016/j.scitotenv.2019.06.498
74  Li, X., Wang, Y., Cui, J., Shi, Y., Cai, Y., 2024. Occurrence and Fate of Per- and Polyfluoroalkyl
75      Substances (PFAS) in Atmosphere: Size-Dependent Gas-Particle Partitioning, Precipitation

Scavenging, and Amplification. Envion. Sci. Technol. 58, 9283-9291. http://dx.doi.org/10.1021/acs.est.4c00569

McMurdo, C.J., Ellis, D.A., Webster, E., Butler, J., Christensen, R.D., Reid, L.K., 2008. Aerosol enrichment of the surfactant PFO and mediation of the water - Air transport of gaseous PFOA. Envion. Sci. Technol. 42, 3969-3974. http://dx.doi.org/10.1021/es7032026

Turpin, B.J., Huntzicker, J.J., Hering, S.V., 1994. Investigation of organic aerosol sampling artifacts in the los angeles basin. Atmos. Environ. 28, 3061-3071. http://dx.doi.org/10.1016/1352-2310(94)00133-6

Wu, J., Jin, H., Li, L., Zhai, Z., Martin, W., Hu, J., et al., 2019. Atmospheric perfluoroalkyl acid occurrence and isomer profiles in Beijing, China. Environ. Pollut. 225. http://dx.doi.org/10.1016/j.envpol.2019.113129

**Comment 3:** L.126-130 – The sentences are not clear. Please rephrase.

Response: The sentences have been rephrased.

Lines 124 – 133 (New Version): Before sampling, quartz filters were wrapped in aluminum foil and baked in a muffle furnace at 450°C for 5 hours to eliminate organic components. They were then placed in a super clean room (temperature of $20 \pm 5°C$; relative humidity of $50 \pm 5\%$) for 48 hours. Clean the instrument with alcohol cotton before and after each sampling and record the standard state volume of the sampler. Quartz filters were weighed twice before and after sampling respectively, and the error between the two weighing was not more than 10 mg. After weighing the quartz filter, the quartz filter was wrapped in aluminum foil and stored at $-18°C$. The above experimental processes were carried out in the ultra-clean room.

**Comment 4:** Section 2.2 – Please provide more information on the product details of the PFAA and mass labelled PFAA mix.

Response: More information of the PFAA and mass labelled PFAA mix has been supplemented in Supplementary Table. S1 and S2.

Table. S1. PFAAs CAS and corresponding internal standard substance

| Compound | CAS | Internal Standard | Relative Molecular Mass | Retention time (min) |
|---|---|---|---|---|
| PFBA | 375-22-4 | $^{13}C_4$PFBA | 214.04 | 2.7 |
| PFPeA | 2706-90-3 | $^{13}C_4$PFBA | 264.05 | 3.9 |
| PFHxA | 307-24-4 | $^{13}C_4$PFHxA | 314.06 | 5.1 |
| PFHpA | 375-85-9 | $^{13}C_4$PFHxA | 364.07 | 5.4 |

Continued Table. S1

| Compound | CAS | Internal Standard | Relative Molecular Mass | Retention time (min) |
|---|---|---|---|---|
| PFOA | 335-67-1 | $^{13}C_4$PFOA | 414.08 | 6.1 |
| PFNA | 375-95-1 | $^{13}C_4$PFNA | 464.09 | 6.9 |
| PFDA | 335-76-2 | $^{13}C_4$PFDA | 514.10 | 7.5 |
| PFUnDA | 2058-94-8 | $^{13}C_4$PFUnDA | 564.11 | 7.8 |
| PFDoDA | 307-55-1 | $^{13}C_2$PFDoDA | 614.12 | 8.6 |
| PFTrDA | 72629-94-8 | $^{13}C_2$PFDoDA | 664.13 | 9.2 |
| PFTeDA | 376-06-7 | $^{13}C_2$PFDoDA | 714.14 | 9.4 |
| PFHxDA | 67905-19-5 | $^{13}C_2$PFDoDA | 814.16 | 10.2 |
| PFODA | 16517-11-6 | $^{13}C_2$PFDoDA | 914.18 | 10.8 |
| PFBS | 375-73-5 | $^{18}O_2$PFHxS | 300.11 | 11.0 |
| PFHxS | 355-46-4 | $^{18}O_2$PFHxS | 400.14 | 11.8 |
| PFOS | 1763-23-1 | $^{13}C_4$PFOS | 500.16 | 13.2 |
| PFDS | 335-77-3 | $^{13}C_4$PFOS | 600.18 | 14.4 |
| $^{13}C_4$PFBA | | | 226.04 | 2.7 |
| $^{13}C_4$PFHxA | | | 326.04 | 5.1 |
| $^{13}C_4$PFOA | | | 426.05 | 6.9 |
| $^{13}C_4$PFNA | | | 476.06 | 7.5 |
| $^{13}C_4$PFDA | | | 526.07 | 7.8 |
| $^{13}C_4$PFUnDA | | | 576.08 | 8.6 |
| $^{13}C_2$PFDoDA | | | 626.09 | 9.2 |
| $^{18}O_2$PFHxS | | | 402.10 | 9.4 |
| $^{13}C_4$PFOS | | | 526.08 | 10.2 |

105

Table. S2. PFAAs standard and corresponding internal standard substances and test information

| Compound | Internal Standard | Standard (internal standard) Precursor Ion (m/z) | Standard (internal standard) Product Ion (m/z) | Standard (internal standard) DP (V) | Standard (internal standard) CE (V) | Mark recovery(%) | MDL (ng·L$^{-1}$) |
|---|---|---|---|---|---|---|---|
| PFBA | $^{13}$C$_4$PFBA | 213(217) | 169(172) | –40(–50) | –13(–12) | 97.49–112.02 | 0.3 |
| PFPeA | $^{13}$C$_4$PFBA | 263(217) | 219/69(172) | –40(–50) | –10/–50(–12) | 73.61–112.98 | 0.2 |
| PFHxA | $^{13}$C$_4$PFHxA | 313(315) | 269/119(270) | –45(–55) | –13/–27(–14) | 94.84–115.89 | 0.2 |
| PFHpA | $^{13}$C$_4$PFHxA | 363(315) | 319/169(270) | –30(–55) | –14/–24(–14) | 71.74–111.84 | 0.2 |
| PFOA | $^{13}$C$_4$PFOA | 413(417) | 369/169(372) | –40(–70) | –14/–24(–20) | 91.04–117.75 | 0.3 |
| PFNA | $^{13}$C$_4$PFNA | 463(468) | 419/169(423) | –35(–70) | –16/–24(–22) | 92.55–112.96 | 0.2 |
| PFDA | $^{13}$C$_4$PFDA | 513(515) | 469/219(470) | –40(–75) | –18/–26(–17) | 96.81–115.60 | 0.2 |
| PFUnDA | $^{13}$C$_4$PFUnDA | 563(565) | 519/319(520) | –70(–60) | –16/–28(–15) | 96.81–115.24 | 0.2 |
| PFDoDA | $^{13}$C$_2$PFDoDA | 613(615) | 569/169(570) | –70(–60) | –18/–36(–15) | 97.46–116.71 | 0.2 |
| PFTrDA | $^{13}$C$_2$PFDoDA | 663(615) | 619/169(570) | –65(–60) | –20/–38(–15) | 96.88–110.99 | 0.3 |
| PFTeDA | $^{13}$C$_2$PFDoDA | 713(615) | 669/169(570) | –85(–60) | –20/–38(–15) | 98.10–113.01 | 0.2 |
| PFHxDA | $^{13}$C$_2$PFDoDA | 813(615) | 769/169(570) | –90(–60) | –18/–30(–15) | 99.38–118.08 | 0.3 |
| PFODA | $^{13}$C$_2$PFDoDA | 913(615) | 869/169(570) | –40(–60) | –25/–45(–15) | 85.64–104.97 | 0.2 |
| PFBS | $^{18}$O$_2$PFHxS | 299(403) | 80/99(103) | –90(–90) | –70/–38(–75) | 71.27–106.25 | 0.3 |
| PFHxS | $^{18}$O$_2$PFHxS | 399(403) | 80/99(103) | –90(–90) | –90/–72(–75) | 89.91–102.78 | 0.3 |
| PFOS | $^{13}$C$_4$PFOS | 499(503) | 80/99(80) | –105(–90) | –110/–98(–95) | 96.42–111.07 | 0.3 |
| PFDS | $^{13}$C$_4$PFOS | 599(503) | 80/99(80) | –120(–90) | –124/–110(–95) | 97.56–109.07 | 0.2 |

**Comment 5:** Section 2.4 – Were matrix effects evaluated?

Response: We sincerely appreciate the reviewer for raising this critical point. In this study, two field blanks and procedure blanks were included during each sampling period. Notably, the PFAA concentrations detected in both field and procedure blanks were below the MDLs. These results indicated that the influence of matrix effects exerted on experimental results was negligible, thereby proving the reliability of the experimental data.

**Comment 6:** L.174 – I suggest reporting procedure blank values of the targeted PFAS in the SI.

Response: As addressed in response to **Comment 5**, PFAA concentrations in procedure blanks were below the MDLs, and the information has been added in line 177 in this manuscript.

Line 178: PFAAs were not detected in field blanks and program blanks.

**Comment 7:** L.192 – The authors report a maximum average seasonal PFAA concentration of 181.63 pg/m³. However, in Fig. 1, the average concentrations of PFBA, PFOA, and PFOS exceed 200 pg/m³. Could the authors clarify this discrepancy?

Response: The data in the manuscript (46.68 – 181.63 pg·m–3) refers to the average PFAA concentrations for each season and the data in Figure 1 refers to the PFAA concentrations for the four seasons. These sentences have been rephrased in this manuscript.

Lines 193 – 201 (New Version): The PFAA average concentrations ranged from 46.68 to 181.63 pg·m$^{-3}$ in Fig. 1 across four seasons. However, the increased airflow during pump operation enhanced the adsorption of gaseous PFAA on quartz filters (Turpin et al., 1994; McMurdo et al., 2008; Ahrens et al., 2012; Chang et al., 2024), which may lead to a slight overestimated of PFAA values in this study. The PFAA average concentrations were comparable to levels observed in Chengdu (150 pg·m$^{-3}$) (Fang et al., 2019), but significantly higher than those recorded in Shenzhen (8.80 pg·m$^{-3}$) (Liu et al., 2015a) and the average concentration in China (39.84 pg·m$^{-3}$) (Han et al., 2019).

Line 215 (New Version): Fig. 1. Box diagram of 17 PFAA concentrations in PM$_{2.5}$ across four seasons.

Reference:

Fang, S., Li, C., Zhu, L., Yin, H., Yang, Y., Ye, Z., et al., 2019. Spatiotemporal distribution and isomer profiles of perfluoroalkyl acids in airborne particulate matter in Chengdu City, China. Sci. Total. Environ. 689, 1235-1243. http://dx.doi.org/10.1016/j.scitotenv.2019.06.498

Han, D., Ma, Y., Huang, C., Zhang, X., Xu, H., Zhou, Y., et al., 2019. Occurrence and source apportionment of perfluoroalkyl acids (PFAAs) in the atmosphere in China. Atmos. Chem. Phys. 19, 14107-14117. http://dx.doi.org/10.5194/acp-19-14107-2019

Liu, B., Zhang, H., Yao, D., Li, J., Xie, L., Wang, X., et al., 2015a. Perfluorinated compounds (PFCs) in the atmosphere of Shenzhen, China: Spatial distribution, sources and health risk assessment. Chemos. 138, 511-518. http://dx.doi.org/10.1016/j.chemosphere.2015.07.012

**Comment 8:** L.206, 210, and 221 – I assume the primary substitutes of PFOA and PFOS are the compounds mentioned in lines 204 and 205. However, this is difficult to understand. Please clearly mention which compounds are the substitutes for PFOA and PFOS.

Response: The substitutes of PFOA and PFOS have been added in this manuscript.

Lines 210 – 214 (New Version): During the study period, PFOA and PFOS along with its primary substitutes (PFOA primary substitutes: PFBA and PFHxA. PFOS primary substitutes: PFPeA and PFBS.) accounted for 23%–34% and 18.1%–29.9% of total PFAAs, consistent with the research (Liu et al., 2017).

Reference:

Liu, Z., Lu, Y., Wang, P., Wang, T., Liu, S., Johnson, A.C., et al., 2017. Pollution pathways and release estimation of perfluorooctane sulfonate (PFOS) and perfluorooctanoic acid (PFOA) in central and eastern China. Sci. Total. Environ. 580, 1247-1256. http://dx.doi.org/10.1016/j.scitotenv.2016.12.085

**Comment 9:** Fig. 2- Please provide the unit in the x-axis of the second plot. Also, label the plots as 'a' and 'b'.

Response: The x-axis unit has been explicitly labeled in Fig. 2, with plots clearly labelled using 'a' and 'b'.

[Figure]

Fig. 2. PFAA concentrations characteristics across four seasons

**Comment 10:** L.237-239 – What were the different types of chemical industries near the sampling region? Were any fluorochemical manufacturing plants present within the vicinity of the sampling region?

Response: The chemical industrial profile of the study area primarily encompasses rubber manufacturing, fine chemicals production, pharmaceutical intermediates synthesis, and advanced materials development. These industries are also associated with emissions of fluorinated products. For example, Alchemist-pharm Chemical Technology Co., Ltd. (http://www.alchemist-pharm.com/En), situated 2.8 km distance from the study area, can produce fluorine-containing chemicals, and is a representative presence in the region's fluorine chemical manufacturing.

**Comment 11:** L.247-249 – This sentence is unclear. What do the concentrations (0.26–1.90 pg/m³) in parentheses represent? Do they indicate the range of total PM$_{2.5}$ PFAA concentrations in the Middle-Lower Yangtze River plains? Were these values measured in this study, or are they sourced from the literature? If they are from the literature, please cite the original source instead of referencing the critical review by

Faust (2023).

Response: The concentrations (0.26–1.90 pg·m$^{-3}$) in parentheses represented PFAA concentrations in the Middle-Lower Yangtze River plains. The values sourced from the literature. The reference format has been modified.

Lines 255 – 257 (New Version): The air mass originated from Hubei Province, passed through Middle-Lower Yangtze River plains (PFAA concentrations: 0.26–1.90 pg·m$^{-3}$) (Faust et al., 2023), and then entered the study region.

Reference:

Faust, J.A., 2023. PFAS on atmospheric aerosol particles: a review. Environ. Sci.-Process Impacts 25, 133-150. http://dx.doi.org/10.1039/d2em00002d

**Comment 12:** L.281-290 – Were there any wastewater treatment plants (WWTPs) in the vicinity of the study region? WWTPs are also reported to introduce PFAS into the atmosphere through aerosolisation and volatilisation during treatment processes such as aeration (please refer to the studies by Qiao et al. 2024 (https://doi.org/10.1016/j.jhazmat.2024.134879), Lin et al. 2022 (https://doi.org/10.1016/j.envint.2022.107434)).

Response: The WWTP of near the study area is Zhongyuan Environmental Protection Wulongkou Water Affairs Branch Company (distance: 7.8 km, total designed daily treatment capacity: 200,000 m$^{-3}$day$^{-1}$, http://www.cpepgc.com/20180626/77.html). Other WWTPs are located farther away, such as Chen Sanqiao WWTP (distance: 7.8 km, total designed daily treatment capacity: 150,000 m$^{-3}$day$^{-1}$, https://public.zhengzhou.gov.cn/D250406X/196731.jhtml).

**Comment 13:** 291- Please provide the correct reference.

Response: The references have been revised.

Lines 298 – 299 (New Version): This result was consistent with conclusions drawn by Chen et al. (2021) and Han et al. (2019).

Reference:

Chen, M., Wang, C., Gao, K., Wang, X., Fu, J., Gong, P., et al., 2021. Perfluoroalkyl substances in precipitation from the Tibetan Plateau during monsoon season: Concentrations, source regions and mass fluxes. Chemos. 282. http://dx.doi.org/10.1016/j.chemosphere.2021.131105

Han, D., Ma, Y., Huang, C., Zhang, X., Xu, H., Zhou, Y., et al., 2019. Occurrence and source apportionment of perfluoroalkyl acids (PFAAs) in the atmosphere in China. Atmos. Chem. Phys. 19, 14107-14117. http://dx.doi.org/10.5194/acp-19-14107-2019

**Comment 14:** L.312 – Could other precursor compounds such as polyfluoroalkyl phosphate esters (PAP) or perfluorooctane sulfonamides (FOSA) degrade into these compounds?

Response: PAP are a class of PFAS. These substances typically exhibit a telomer-based chemical structure (denoted as n:2 PAP, such as 6:2 and 8:2 PAP), with their primary degradation products being short-chain perfluorocarboxylic acids (PFCAs). Theoretically, PAP with longer telomeric chains (e.g., 11:2 and 12:2 PAP) could degrade into long-chain PFAAs. However, in practical applications, the industrial use of long-chain PAP ($\geq$C11) remains limited, and environmental studies on their behavior and fate are sparse. Current scientific literature predominantly focuses on telomer-based PAPswithin the C6 – C10 chain-length range. FOSA features a fixed C8 carbon chain structure and its principal degradation end-product is PFOS.

Lines 324 – 326 (New Version): Long-chain PFAAs (C11–C14) were known degradation products of Long-chain FTOHs (Liu et al., 2017; Thackray and Selin, 2017; Wang et al., 2014).

Reference:

Liu, Z., Lu, Y., Wang, P., Wang, T., Liu, S., Johnson, A.C., et al., 2017. Pollution pathways and release estimation of perfluorooctane sulfonate (PFOS) and perfluorooctanoic acid (PFOA) in central and eastern China. Sci. Total. Environ. 580, 1247-1256. http://dx.doi.org/10.1016/j.scitotenv.2016.12.085

Thackray, C.P., Selin, N.E., 2017. Uncertainty and variability in atmospheric formation of PFCAs from fluorotelomer precursors. Atmos. Chem. Phys. 17, 4585-4597. http://dx.doi.org/10.5194/acp-17-4585-2017

Wang, Y. Q., 2014. Meteolnfo: GIS software for meteorological data visualization and analysis. Meteorol. Appl. 21, 360-368. https://doi.org/10.1002/met.1345

**Comment 15:** Fig. 5 – the caption for Figure 5(C) is missing, please include this.

Response: The caption for Figure 5(C) have been added.

Line 332 – 333 (New Version): Fig. 5. The source distribution spectrum of PFAAs in PMF (a), the annual source proportion diagram (b) and the winter source proportion diagram (c)

**Technical corrections:**

**Comment 1:** L.48-49 – The phrasing of the sentence is a bit awkward. Please rephrase.

Response: The sentence has been rephrased.

Lines 47 – 49 (New Version): PFAAs levels in the atmosphere have attracted adequate attention due to the bioaccumulation and potential toxicity of PFAAs.

**Comment 2:** L.77 and 79 – please provide the in-text citation in the correct format.

Response: The format has been revised.

Lines 77 – 82 (New Version): Han et al. (2022) employed positive matrix factorization (PMF) to identify four sources of PFAAs within the atmosphere. Meanwhile, Chen et al. (2021) and Wang et al. (2022b) combined principal component analysis with back-trajectory model to assess air mass influence PFAA concentrations in precipitation from the Tibetan Plateau and airborne particulate matter in Chengdu, China.

Reference:

Chen, M., Wang, C., Gao, K., Wang, X., Fu, J., Gong, P., et al., 2021. Perfluoroalkyl substances in precipitation from the Tibetan Plateau during monsoon season: Concentrations, source regions and mass fluxes. Chemos. 282. http://dx.doi.org/10.1016/j.chemosphere.2021.131105

Han, D., Ma, Y., Huang, C., Zhang, X., Xu, H., Zhou, Y., et al., 2019. Occurrence and source apportionment of perfluoroalkyl acids (PFAAs) in the atmosphere in China. Atmos. Chem. Phys. 19, 14107-14117. http://dx.doi.org/10.5194/acp-19-14107-2019

Wang, S., Lin, X., Li, Q., Liu, C., Li, Y., Wang, X., 2022b. Neutral and ionizable per-and polyfluoroalkyl substances in the urban atmosphere: Occurrence, sources and transport. Sci. Total. Environ. 823. http://dx.doi.org/10.1016/j.scitotenv.2022.153794

**Comment 3:** L.150 – Change the word 'extracts were' to 'extraction was'.

Response: The sentence has been revised.

Lines 151 – 153 (New Version): After the addition of methanol, the extraction was performed 3 times by sonication. Following the centrifugation (4500 r/min, 15 min), the extracts were diluted with ultrapure water.

**Comment 4:** L.156 and 163 – Supplementary section 1.2 details the source apportionment analysis. The instrument analysis is detailed in section 1.1 of the SI. Please change this.

Response: The 'Supplementary 1.2' in section 2.3 have been changed.

Lines 156 – 157 (New Version): Detailed steps for sample pretreatment are documented in Supplementary 1.1.1.

Lines 163 – 164 (New Version): Comprehensive details regarding the instrumental analysis can be found in Supplementary 1.1.2.

**Comment 5:** L.41 of SI – The 'Evaporationoff' seems to be a typo. Please correct this.

Response: The word has been revised in this supplementary.

Lines 40 – 42 (New Version): Nitrogen Evaporation was performed using a nitrogen evaporator to completely dry the eluate (the nitrogen blow temperature should not exceed 40°C, and no bubbles should be present on the liquid surface).

---

## Referee Report (RR1)

Thank you for addressing my previous comments carefully. I understand the authors tested for gaseous-phase PFAS adsorption using a Teflon—Quartz double filter system, following Turpin et al. (1994). The authors reported that PFAS were below MDL level in the second (quartz) filter and thus suggest that adsorption artefacts from gaseous PFAS are negligible. While I appreciate these efforts, I have some follow-up questions and suggestions that I believe should be addressed before final acceptance:

- 1. Where was the double filter system experiment conducted? Was it carried out at the same location and under similar environmental conditions as the main PM sampling campaign? The modified manuscript text does not provide this important contextual information.
- 2. The added text (Lines 184–189) could be misinterpreted. It gives the impression that the double filter system was employed throughout the study, rather than being used in a one-time supplementary experiment.
- 3. From the response, it appears that only a single experiment with the double filter setup was performed. Could the detection of PFAS in the quartz filter below MDL levels simply be due to low atmospheric PFAS levels on that specific day?
- 4. Since the double filter system was not employed during the main sampling campaign, the potential for gaseous PFAS adsorption onto quartz fiber filters remains a valid concern. While your additional test is useful, it does not fully rule out the occurrence of positive sampling artefacts throughout the study period.

---

## Author Response (AR2)

**Itemized Response to Reviewer's Comments**

**Ms. Ref. No.**: egusphere-2024-4147

Title: Characteristics, main sources, health risks of PM2.5-bound perfluoroalkyl acids

in Zhengzhou, central China: From seasonal variation perspective

We sincerely apologize for our response so late. This is because we conducted the necessary experiments and consulted many papers to comprehensively solve the positive sampling artefacts. After experimental verification, the influence of positive sampling artefacts in this study can be ignored. We sincerely thank the editor and reviewer once again for their valuable and constructive comments on our manuscript. We have carefully addressed all the comments and believe that the quality of the manuscript has been further enhanced through this round of revisions. All modifications made in this revision are highlighted in red in the revised manuscript for ease of reference.

**General comments:**

Thank you for addressing my previous comments (Reviewer 2) in the revised manuscript. While I appreciate the efforts made, I believe there is still room for improvement in terms of the overall readability and clarity of the manuscript. Below, I provide further comments in response to the authors' answers to my earlier questions.

Response: Thank you again for your valuable and constructive comments on our manuscript. Detailed revisions have been listed below.

**Specific comments:**

Response to comment 2 - specific comments. The author's response to the comment on positive sampling artefacts is somewhat unclear. Specifically, I don't understand the link between positive sampling artefacts and practices like filter baking or the observation that PFAA levels were below MDL in process blanks, as this does not directly address the issue. As previously noted, certain PFAS compounds are known to partition between gas and particle phases. Given that several ionic PFAS are semi-volatile, they may exist in the gas phase and can adsorb onto quartz fiber filters during PM sampling. While the authors briefly mention the adsorption of gaseous PFAS in the revised manuscript, this discussion remains insufficient. It should be also noted that positive sampling artefacts such as adsorption of gaseous PFAS along with PFAS in particulate phase are not only due to increased flow rate but also could be due to the physicochemical properties of PFAS and filter material. I think the discussion on positive sampling artefacts is not adequate and a slightly better discussion should be given.

Response: We sincerely apologize for the delayed response to this comment. This was due to the implementation of additional experiments. Upon careful consideration of the reviewer's comment, we recognized that our study initially lacked an assessment of positive sampling artefacts. To address this gap, we conducted supplementary experiments to quantify the adsorption of gaseous-phase PFAAs onto quartz filters. We used the Teflon filter for particulate matter filtration and the quartz filter for the adsorption of gaseous-phase PFAAs. PFAA levels were below the MDL in the quartz filter sample. Furthermore, we have also discussed this comment in the

manuscript (Lines 184 - 189) in the new version.

Quartz filters are routinely employed for the collection of atmospheric organic carbon. Pre-combustion effectively eliminates interference from inherent organic background material within quartz filters (Kirchstetter et al., 2001), ensuring the accuracy of sample concentration measurements. Furthermore, baked quartz filters demonstrate low affinity for adsorption of organic compounds (Jung et al., 2011).

As the reviewer rightly points out, PFAAs may exist in both the gaseous and particulate phases. Positive sampling artefacts are a recognized phenomenon in such measurements (Turpin et al., 1994 and Chang et al., 2024), and it is acknowledged that the concentrations reported in this study have the potential to be overestimated due to this effect. To directly address this important comment, we implemented additional experiments, the details and results of which are presented below. To specifically isolate gaseous-phase PFAAs, we used a dual-filter sampling system: The Teflon filter was positioned upstream to remove particulate matter, followed by a quartz filter downstream to capture gas-phase PFAAs adsorbed onto the quartz filter (Turpin et al., 1994). The sampling period was from 10:00 to 9:00 the next day on June 10, 2025 (23 hours). Sample pretreatment, comprising ultrasonic extraction followed by solid-phase extraction, was performed prior to analysis. Two distinct samples were analyzed through UPLC-MS/MS: a 0.0001 μg·mL-1 PFAAs standard solution and the quartz filter sample. As shown in Fig. 1b, no detectable chromatographic peaks were observed in the quartz filter sample, indicating that PFAA levels were below the MDL. This result demonstrated that the impact of positive sampling artefacts on our findings could be ignored.

Lines 184 – 189 (New Version): This study used a dual-filter sampling system: The Teflon filter was positioned upstream to remove particulate matter, followed by a quartz filter downstream to capture gas-phase PFAAs adsorbed onto the quartz filter (Turpin et al., 1994). PFAA levels were below the MDL in the quartz filter sample. This result indicated that the impact of positive sampling artefacts in this study could be ignored.

Fig.1 Chromatograms of 0.0001  $\mu$ g·mL-1 PFAAs standard solution (a) and quartz filter sample (b) Reference:

Chang, NY., Eichler, CMA., Amparo, DE., Zhou, JQ., Baumann, K., Hubal, EAC., et al., 2024. Indoor air concentrations of PM2.5 quartz fiber filter-collected ionic PFAS and emissions to outdoor air: findings from the IPA campaign. Environ. Sci.-Process Impacts. 28, 3061-3071. https://dx.doi.org/10.1039/D4EM00359D

Jung, J., Kim, YJ., Lee, KY., Kawamura, K., Hu, M., Kondo, Y., 2001. The effects of accumulated refractory particles and the peak inert mode temperature on semi-continuous organic carbon and elemental carbon measurements during the CAREBeijing 2006 campaign. Atmos. Environ. 45, 7192-7200. https://dx.doi.org/10.1016/j.atmosenv.2011.09.003

Kirchstetter, TW., Corrigan, CE., Novakov, T., 2001. Laboratory and field investigation of the adsorption of gaseous organic compounds onto quartz filters. Atmos. Environ. 35, 1663-1671. https://dx.doi.org/10.1016/S1352-2310(00)00448-9

Turpin, BJ., Huntzicker, JJ., Hering, SV., 1994. Investigation of organic aerosol sampling artifacts in the los angeles basin. Atmos. Environ. 28, 3061-3071. https://dx.doi.org/10.1016/1352-2310(94)00133-6

Response to comment 5 - specific comments

Matrix effects are about suppression/enhancement of ionisation of analytes due to the presence of matrix, which can still occur even if your blank is "clean". A blank below MDL values could mean the matrix doesn't contain detectable levels of your analyte, but it doesn't confirm that the matrix won't affect the analyte's signal when it is present.

ESI is shown to be prone to matrix effects, leading to ion suppression or enhancement due to the interference of certain inorganic salts and compounds (Chekmeneva et al., 2017 – https://doi.org/10.1021/acs.jproteome.6b01003; Silva et al., 2016 - https://doi.org/10.5935/0103-5053.20150296). Atmospheric PM constituents such as sulfates, nitrates, and ammonium salts could cause ion suppression in ESI (Kourtchev et al., 2020 - https://doi.org/10.1021/acs.analchem.0c00971). Therefore, it is important to assess the matrix effect even if you have various blanks.

Response: As rightly noted, the determination is often subject to matrixeffect due to the presence of matrix components coeluting with the analyte of interest during LC-MS. The matrix effect-introduced MS signal suppression or enhancement may lead to erroneous results. (Smeraglia et al., 2002). To avoid the influence of matrix effects in this study, Solid Phase Extraction (SPE) was employed for sample pretreatment to enabled the selectively extract PFAAs from the sample (Cao et al., 2015 and de Navarro et al., 2024). Specifically, weak anion exchange (WAX) cartridges were utilized for the purification of PFAAs. The WAX cartridge, incorporating polar, reversed-phase, and ion-exchange functional groups, interacts effectively with the acidic functional groups characteristic of PFAAs, facilitating their retention (Hennrich et al., 2011 and Cao et al., 2024). The acidic functional groups present in PFAAs structures enable this specific interaction with the functional groups on the WAX sorbent, ensuring their capture on the extraction cartridge (Zou et al., 2024). The SPE procedure involved activation of the WAX cartridge, enrichment of PFAAs, washing to remove non-specific impurities (such as components like sulfate, nitrate, ammonium salt, etc), and elution of the target compounds. This

comprehensive process achieved both enrichment and purification of the PFAAs, thereby effectively eliminating the interference from sample matrix components.

Recovery has been widely used to evaluate the accuracy and reliability of analytical methods for PFAAs (Fang, 2019 and Han, 2019). Recovery refers to the ratio of the result obtained by LC-MS analysis to the theoretical value after adding a known amount of standard material to the sample. In this study, the recovery rates ranged from 71.27%–118.08%.

$$R = \frac{Q_1 - Q_2}{M} \times 100\%$$

Where:

R is the recovery rate,

Q1 is the measured concentration in the methanol sample spiked with the standard,

Q2 is the measured concentration in the blank methanol matrix,

M is the concentration of the added standard.

**Reference:**

- Cao, FM., Wang, L., Sun, HW., Yang, J., Wang, RN., 2015. The optimization of sorbents and elution method for perfluorocarboxylic acids from the aquatic solution. J. Am. Fresenius Environ. Bull. 24, 2238-2244. https://orcid.org/0000-0002-8193-9954
- Cao, WK., Chu, PY., Bruening, ML., Shi, RL., Hernandez-Barry, H., Tran, JC., 2024. Efficient, Low-Cost, and High-Throughput Sodium Dodecyl Sulfate (SDS) Removal from Protein Digests Using Weak-Anion Exchange. J. Am. Soc. Mass Spectrom. 36, 680-687. http://dx.doi.org/10.1021/jasms.4c00304
- de Navarro, MG., Reyna, Y., Quinete, N., 2024. It's raining PFAS in South Florida: Occurrence of per- and polyfluoroalkyl substances (PFAS) in wet atmospheric deposition from Miami-Dade, South Florida. J. Am. Atmos. Pollut. Res. 15, 102302. http://dx.doi.org/10.1016/j.apr.2024.102302
- Fang, S., Li, C., Zhu, L., Yin, H., Yang, Y., Ye, Z., et al., 2019. Spatiotemporal distribution and isomer profiles of perfluoroalkyl acids in airborne particulate matter in Chengdu City, China. Sci. Total. Environ. 689, 1235-1243. <a href="http://dx.doi.org/10.1016/j.scitotenv.2019.06.498">http://dx.doi.org/10.1016/j.scitotenv.2019.06.498</a>
- Han, D., Ma, Y., Huang, C., Zhang, X., Xu, H., Zhou, Y., et al., 2019. Occurrence and source apportionment of perfluoroalkyl acids (PFAAs) in the atmosphere in China. Atmos. Chem. Phys. 19, 14107-14117. http://dx.doi.org/10.5194/acp-19-14107-2019
- Hennrich, ML., Groenewold, V., Kops, GJPL., Heck, AJR., Mohammed, S., 2011. Improving Depth in Phosphoproteomics by Using a Strong Cation Exchange-Weak Anion Exchange-Reversed Phase Multidimensional Separation Approach. Anal. Chem. 83, 7137-7143. http://dx.doi.org/10.1021/ac2015068

- Smeraglia, J., Baldrey, SF., Watson, D., 2002. Matrix effects and selectivity issues in LC-MS-MS. Chromatographia. 55, S95-S99. http://dx.doi.org/10.1007/BF02493363
- Zou, JM., Zhao, MZ., Chan, SA., Song, Y., Yan, SW., Song, WH., Song, et al., 2024. Rapid and simultaneous determination of ultrashort-, short- and long- chain perfluoroalkyl substances by a novel liquid chromatography mass spectrometry method. J. Chromatogr. A. 1734, 465324. http://dx.doi.org/10.1016/j.chroma.2024.465324

Response to comment 11 - specific comments

Thank you for clarifying that the concentrations (0.26–1.90 pg·m-3) refer to PFAA levels in the Middle–Lower Yangtze River plains. Since these values are sourced from original studies, it would be more appropriate to cite the original research article rather than the critical review by Faust et al. (2023), as previously suggested.

Response: We have modified it to the critical review.

Lines 260 − 262 (New Version): The air mass originated from Middle-Lower Yangtze River plains (PFAA concentrations: 0.26–1.90 pg·m-3) (Lu et al., 2018) and then entered the study region from Hubei Province.

**Reference:**

Lu, Z., Lu, R., Zheng, H., Yan, J., Song, L., Wang, J., et al., 2018. Risk exposure assessment of per- and polyfluoroalkyl substances (PFASs) in drinking water and atmosphere in central eastern China. Environ. Sci. Pollut. Res. 25, 9311-9320. http://dx.doi.org/10.1007/s11356-017-0950-x

---

## Author Response (AR3)

**Itemized Response to Reviewer's Comments**

**Ms. Ref. No.**: egusphere-2024-4147

**Title:** Characteristics, main sources, health risks of PM2.5-bound perfluoroalkyl acids in Zhengzhou, central China: From seasonal variation perspective

We have carefully addressed your comments on our manuscript and made necessary revisions of the previous manuscript. We sincerely thank you for valuable and constructive inputs. We believe that we have adequately addressed all of your comments and thus the current version has been greatly improved with those valuable comments and further English editing. All modifications made in this revision are highlighted in red in the revised manuscript for ease of reference.

**General comments:**

Thank you for addressing my previous comments carefully. I understand the authors tested for gaseous-phase PFAS adsorption using a Teflon–Quartz double filter system, following Turpin et al. (1994). The authors reported that PFAS were below MDL level in the second (quartz) filter and thus suggest that adsorption artefacts from gaseous PFAS are negligible. While I appreciate these efforts, I have some follow-up questions and suggestions that I believe should be addressed before final acceptance.

**Comment 1:** Where was the double filter system experiment conducted? Was it carried out at the same location and under similar environmental conditions as the main PM sampling campaign? The modified manuscript text does not provide this important contextual information.

Response: We apologize for the insufficient explanation in our previous response, which has led to this question of yours. The double filter system experiment was conducted at the same location as the main PM2.5 sampling campaign, which was on the rooftop of the Collaborative Innovation Building at Zhengzhou University (34°48′N, 113°31′E), with a height of 14 meters. This location is approximately 500 meters east of the West Fourth Ring Road and 2 kilometers south of the Lianhuo Expressway.

The experiment was carried out under the same environmental conditions as the main sampling campaign. The sampling conditions, such as the use of quartz filters (pretreated by baking and conditioning in a super clean room), the sampler flow rate (100 L·min-1), and the avoidance of adverse weather conditions (rain, snow, power outages) that would invalidate samples, were consistent for both the main PM2.5 sampling and the double filter system experiment. Additionally, strict quality control measures (e.g., avoiding fluorinated plastic materials) were applied uniformly across all sampling and analytical processes, ensuring that the double filter system experiment was integrated into the main campaign's contextual and environmental setup.

We have also supplemented this background information in the original manuscript; please refer to Lines 187 – 190 for details.

Lines 187 - 190 (New Version): The dual-filter sampling system was used for supplementary experiments. The sampling location and conditions were consistent with the main  $PM_{2.5}$  sampling work, and the sampling time was from 10:00 on June 10,2025 to 9:00 on the following day.

**Comment 2:** The added text (Lines 184–189) could be misinterpreted. It gives the impression that the double filter system was employed throughout the study, rather than being used in a one-time supplementary experiment.

Response: Thank you for pointing out this issue. We have supplemented this information in the manuscript, which can be found in Lines 187-190.

Lines 187 - 190 (New Version): The dual-filter sampling system was used for supplementary experiments. The sampling location and conditions were consistent with the main  $PM_{2.5}$  sampling work, and the sampling time was from 10:00 on June 10,2025 to 9:00 on the following day.

**Comment 3:** From the response, it appears that only a single experiment with the double filter setup was performed. Could the detection of PFAS in the quartz filter below MDL levels simply be due to low atmospheric PFAS levels on that specific day?

Response: After comprehensive consideration, we conclude that it cannot be attributed to the low level of PFAAs in the atmosphere on that specific day. The reasons are as follows:

- (1): Due to the relatively high temperatures in summer, elevated temperatures can promote the diffusion of PFAAs from the particulate phase to the gas phase, which may result in higher gaseous PFAA concentrations in summer. (Liu et al., 2018 and Li et al., 2024).
- (2): After the filters were baked (450°C, 5h), their adsorption capacity for gaseous PFAAs was significantly reduced (Jung et al., 2011). Each sample had a sampling duration of 23 hours (sampling volume: 3.25 m3), and this process should represent a balance between the adsorption and desorption of gaseous PFAAs.
- (3): The emission sources of PFAAs remained relatively stable over a certain period. The main factor affecting PFAA concentrations is meteorological conditions,

among which temperature plays a key role in regulating the gas-particle partitioning of PFAAs (Han et al., 2019 and Li et al., 2024).

(4): Each sample was collected over a 23-hour period (from 10:00 on the first day to 09:00 on the next day), which was sufficient to represent the average atmospheric conditions over a period of time rather than transient daily fluctuations.

In summary, we believe the results of the supplementary dual-filter experiment conducted in summer are reliable. The detection of PFAAs below the MDL on the quartz filter indicated that the amount of PFAAs adsorbed by the quartz filter was negligible.

**Reference:**

- Jung, J., Kim, YJ., Lee, KY., Kawamura, K., Hu, M., Kondo, Y., 2001. The effects of accumulated refractory particles and the peak inert mode temperature on semi-continuous organic carbon and elemental carbon measurements during the CAREBeijing 2006 campaign. Atmos. Environ. 45, 7192-7200. https://dx.doi.org/10.1016/j.atmosenv.2011.09.003
- Han, D., Ma, Y., Huang, C., Zhang, X., Xu, H., Zhou, Y., et al., 2019. Occurrence and source apportionment of perfluoroalkyl acids (PFAAs) in the atmosphere in China. Atmos. Chem. Phys. 19, 14107-14117. http://dx.doi.org/10.5194/acp-19-14107-2019
- Li, X., Wang, Y., Cui, J., Shi, Y., Cai, Y., 2024. Occurrence and Fate of Per- and Polyfluoroalkyl Substances (PFAS) in Atmosphere: Size-Dependent Gas-Particle Partitioning, Precipitation Scavenging, and Amplification. Envion. Sci. Technol. 58, 9283-9291. http://dx.doi.org/10.1021/acs.est.4c00569
- Liu, W., He, W., Wu, J., Wu, W., Xu, F., 2018. Distribution, partitioning and inhalation exposure of perfluoroalkyl acids (PFAAs) in urban and rural air near Lake Chaohu, China. Environ. Pollut. 243, 143–151. http://dx.doi.org/10.1016/j.envpol.2018.08.052

Comment 4: Since the double filter system was not employed during the main sampling campaign, the potential for gaseous PFAS adsorption onto quartz fiber filters remains a valid concern. While your additional test is useful, it does not fully rule out the occurrence of positive sampling artefacts throughout the study period."

Response: Regarding your concern about the adsorption of gaseous PFAAs by quartz filters, we conducted a supplementary dual-filter experiment in summer. There

may be relatively high content of gaseous PFAAs in summer (Liu et al., 2018 and Li et al., 2024). The result indicated that the PFAA concentrations in the quartz filter were below the MDL.

Several factors support the conclusion that the PFAAs adsorbed by the quartz filter are negligible: first, the high temperatures in summer may lead to relatively high gaseous PFAA levels (Liu et al., 2018 and Li et al., 2024); second, the adsorption capacity of the filter for gaseous PFAAs is significantly reduced after calcination (Jung et al., 2011); third, the emission sources of PFAAs remain relatively stable over a certain period, and meteorological conditions are the main factors affecting the gas-particle partitioning of PFAAs (Han et al., 2019 and Li et al., 2024). Against this background, the detection of PFAAs below the MDL on the quartz filter confirmed that the adsorbed PFAAs were negligible.

The dual-filter experiment provided direct evidence that gas-phase adsorption was not a major issue under the tested conditions (June 2025). However, as you worried, it could not fully rule out the possibility of positive sampling artefacts occurring throughout the entire study period. Therefore, we have added the following limitation of this study in the manuscript (Lines 192 – 194): "Since the dual-filter experiment was not conducted throughout the entire sampling phase, the possibility of positive sampling artefacts could not be completely excluded."

Lines 192 – 194 (New Version): Since the dual-filter experiment was not conducted throughout the entire sampling phase, the possibility of positive sampling artefacts could not be completely excluded.

**Reference:**

- Jung, J., Kim, YJ., Lee, KY., Kawamura, K., Hu, M., Kondo, Y., 2001. The effects of accumulated refractory particles and the peak inert mode temperature on semi-continuous organic carbon and elemental carbon measurements during the CAREBeijing 2006 campaign. Atmos. Environ. 45, 7192-7200. https://dx.doi.org/10.1016/j.atmosenv.2011.09.003
- Han, D., Ma, Y., Huang, C., Zhang, X., Xu, H., Zhou, Y., et al., 2019. Occurrence and source apportionment of perfluoroalkyl acids (PFAAs) in the atmosphere in China. Atmos. Chem. Phys. 19, 14107-14117. http://dx.doi.org/10.5194/acp-19-14107-2019

- Li, X., Wang, Y., Cui, J., Shi, Y., Cai, Y., 2024. Occurrence and Fate of Per- and Polyfluoroalkyl Substances (PFAS) in Atmosphere: Size-Dependent Gas-Particle Partitioning, Precipitation Scavenging, and Amplification. Envion. Sci. Technol. 58, 9283-9291. http://dx.doi.org/10.1021/acs.est.4c00569
- Liu, W., He, W., Wu, J., Wu, W., Xu, F., 2018. Distribution, partitioning and inhalation exposure of perfluoroalkyl acids (PFAAs) in urban and rural air near Lake Chaohu, China. Environ. Pollut. 243, 143–151. http://dx.doi.org/10.1016/j.envpol.2018.08.052